# Red and Green Quantum Dot Color Filter for Full-Color Micro-LED Arrays

**DOI:** 10.3390/mi13040595

**Published:** 2022-04-10

**Authors:** Bingxin Zhao, Qingqian Wang, Depeng Li, Hongcheng Yang, Xue Bai, Shang Li, Pai Liu, Xiaowei Sun

**Affiliations:** 1The Theory Tech. Co., Ltd., Bao’an, Shenzhen 518126, China; 2Shenzhen Key Lab for Advanced Quantum Dot Display and Lighting, Department of Electrical & Electronic Engineering, Southern University of Science and Technology, Shenzhen 518005, China; wangqq3@sustech.edu.cn (Q.W.); 12131037@mail.sustech.edu.cn (D.L.); yanghongcheng820@sina.com (H.Y.); 11749180@mail.sustc.edu.cn (X.B.); lis@mail.sustc.edu.cn (S.L.); liup7@sustech.edu.cn (P.L.)

**Keywords:** quantum dots, display, photolithography, modified QDs, micro-LED

## Abstract

This work demonstrated color-conversion layers of red and green quantum dots color filter for full-color display arrays. Ligands exchange using (3-glycidyloxypropyl) trimethoxysilane with epoxy functional groups to treat QDs in the liquid phase was performed for photolithography use. The combination of ligands of QDs with photo-initiator played a protective role on QDs. Moreover, the pixel size of green QDCF can be reduced to 50 μm, and a high optical density (OD) of 1.2 is realized.

## 1. Introduction

High resolution, color purity, fast response, and efficient luminosity of micro-LED has great potential in displays. In order to meet the current criterion, micro-LED is required to reduce the single-pixel size down to less than 100 μm [1]. Hence, how to achieve full-color display down to microscale has become a significant issue for micro-LED [2,3,4]. Mass transfer and epitaxial growth for fabricating full-color (RGB) micro-LED are difficult. Fortunately, quantum dots (QDs) can absorb blue light to generate color-converted red and green light, making them ideal for full-color micro-LED display [5,6]. To combine QDs with micro-LED, QD patterning is necessary. Inkjet printing has the benefits of simple and rapid patterning of quantum dots [7,8]. However, it has the issue of the large pixel sizes and nonuniform printing, e.g., the coffee ring effect [9,10]. Photolithography is another potential QD patterning technique with micrometer precision.

In preparing photolithography quantum dot color filters (QDCFs), a fully cured cross-linking photoresist is proper to serve as QD matric for its excellent mechanical properties. However, QDs normally possess different surface chemical properties with photoresists, resulting in significant phase segregation when directly mixed QDs with a photoresist [4]. Hence, this paper introduced QDCF using the photolithography method fabricated by epoxied red and green CdSe/ZnS QDs.

## 2. Experiments

### 2.1. Preparation

Unless specified otherwise, chemicals were purchased from Sigma-Aldrich and used without further purification. In our research, ammonia solution (30%) was added drop by drop to CdSe/ZnS QDs (2 mg/mL) at room temperature (RT) to reach a pH value of 8. A molar ratio of 1:1 (3-glycidyloxypropyl) trimethoxysilane in chloroform solution (10%) was added dropwise to the QD suspension under vigorous stirring. After 24 h, the epoxied QDs were separated by centrifugation, washed and purified sequentially with chloroform, ethanol, and methanol, and finally re-dispersed in toluene. 

Pristine and epoxied QDs were mixed separately with SU-8 3050 photoresist at the same concentration. Spin coating of SU8/QDs in toluene (*v*:*v* 1:1) was performed at 500 rpm for 5 s and 1500 rpm for 40 s. Soft baking was performed for 15 min at 120 °C, followed by UV exposure at 365 nm for 15 s (300 mW cm^−2^). Post-exposure baking was performed for 3 min at 100 °C. The samples were developed by immersion in SU-8 developer for 5 min to remove the unexposed areas, followed by rinsing with isopropylamine and drying with nitrogen gas.

### 2.2. Characterization

The PL spectra and quantum yield (QY) were recorded on C11347-11 Quantaurus-QY (Hamamatsu, Japan), the excitation light source -150 W xenon light source was used for PL measurements. The substrate used in the experiments was a blue LED chip. Photolithography was The brand of mask aligner is SUSS MicroTec (Germany). A double-integrating-sphere system measured OD in this work. Scanning electron microscopy (SEM) images were acquired with a field-emission scanning electron microscope (FE-SEM, JEOL JSM-7401F). Optical measurements were performed at RT under ambient conditions.

## 3. Results and Discussion

The typical backlight of the liquid crystal display (LCD) is shown in Figure 1a. As shown in Figure 1a, the yellow phosphor is excited by a blue backlight (~450 nm) and mixes with an unabsorbed blue backlight for full-spectrum light generation, which is then colored through a traditional color filter (CF). The structure of QDCF introduced in this work shows in Figure 1b. As shown in Figure 1b, the red and green colors are converted by red and green QDs directly from blue micro-LED light, and the blue pixel is transparent so that the blue light can be transmitted. Moreover, due to the narrower full width at half maximum (FWHM) of QDs, a large-area color gamut and higher purity can be readily achieved. 

To compare the effects of different solvents of QDs, we studied the roughness and dispersion of solvent-SU8. The chemical compatibility of different solvents with SU-8 photoresist is shown in Appendix A. Among several different solvents: blank sample, oleylamine, toluene, oleic acid, and octane, toluene results in the best chemical compatibility with SU-8 by showing uniform and flat film after spin coating.

However, other solvents do not dissolve well with SU-8. It is hence difficult for further functionalization and dispersing of QD. Hence, we choose toluene as the QD solvent. 

Figure 2a shows the process flow of QDCF. First, a mixture of QDs and photoresist (QD-PR) was spin-coated on the substrate, followed by soft baking to remove the residual solvent and increase the concentration of the photo-initiator. Under irradiation from the mask aligner, the alignment exposure of the mask and substrate was carried out. The chemical environment of QDs is sensitive; hence we chose SU-8 negative photoresist with good transparency and high viscosity. As shown in Figure 2b, the photo-initiator of the irradiated part has a photochemical reaction to form a catalyst, which catalyzes cross-linking reaction or polyreaction [11]. The unexposed QD-PR was removed from the developer, and the remaining part was cured at hard baking.

QDs are extremely sensitive to the chemical environment and thermal treatment during photolithography [4,12]. The most viable strategy for protecting QDs from degrading environments (UV exposure and hard baking) is ligand exchange. QDs diluted in toluene were modified by (3-glycidyloxypropyl) trimethoxysilane, as described in the experiments section. The epoxied QDs with toluene solvent can be sufficiently dispersed in the SU-8 negative epoxy resist. The surface of QD is covered with an epoxy ligand shell, the existence of these epoxy groups facing the chemical and thermal treatment, which protect QDs from degradation [4].

The photoluminescence (PL) emission of the pristine and epoxied QDCF was characterized. Figure 3 shows the PL spectra of QD film before hard baking and patterned QDCF film. Both green and red QDs’ PL peaks have a ~2 nm blue shift. The blue shift is due to the partial etching of the QDs surface by an enormous number of epoxy ligands [13]. The slightly blue shift indicates that the decreased PLQYs in QD films can be attributed to the PL loss by the higher reabsorption of QDs with QD aggregation [14,15]. Moreover, the full width at half maximum (FWHM) of epoxied QD shows barely broadening, indicating reabsorption of green QDs is suppressed by epoxy ligand exchange. However, the QY of the pristine green QDs decreases dramatically after photolithography. Therefore, it can be shown that the ligand exchange does protect QDs from substantial damage. 

In spin-coating, rotation speeds of less than 800 rpm result in >6 µm thickness film; however, it can cause high surface roughness and bad thickness uniformity of the film. Thus, QDCF prepared in this work has an average thickness of 6 μm. The optical density or absorbance of a material is a logarithmic intensity ratio of the light falling upon the material to the light transmitted through the material [16]. Under this condition, the OD of epoxy green QDCF is about 1.2 under a 150 mg/mL concentration. In contrast, OD of pristine green QDCF with the highest concentration (the highest obtained green QD can achieve) is about 0.5. Optical density (OD) in this work was measured by the double-integrating-sphere system. However, this OD is still not enough for micro-LED display applications. An extra high QD concentration or larger film thickness is indispensable to realize a larger OD. Hence, we intend to further study QD-photoresist and hybrid structure in the future.

The QYs of different QDs and QDCF were compared. QDCFs are pixelated QD films after hard baking, as summarized in Table 1. The PLQY of the pristine green QDs is only 45%, while epoxied green QDCF has a high PLQY of 75.6%. High PLQY plays an important role in characterizing illuminations and displays in color-conversion technology-based displays, resulting in higher conversion loss [17]. Under irradiation, the photo-initiator has a photochemical reaction to form an acid as a catalyst. The acid can cause QDs to be quenched, leading to decreased QY. Hence, the PL extraction enhancement is attributed to the epoxy ligand protection since it relieves acid damage on QDs. 

High QD concentration is generally preferable to increase the ratio of the light radiant power from QDs to illumination. However, this can cause heavy reabsorption, resulting in high conversion loss [18,19]. In our case, we achieved high PLQY with high QD concentration without aggregation, indicating the benefits of epoxy ligand exchange, which could improve the dispersion of QDs in a photoresist. Figure 4a is QDCF pixel arrays in a large area with magnificent quality, and the pixel size is 50 μm. As shown in Figure 4a,b, a film thickness of 6 μm of green epoxy QDCF shows a good QD dispersion, further indicating that the epoxy groups connect on the surface of QDs. Compared to pristine QDCF, the emission peaks of epoxied QDCF have a slight blue shift, and the QY increases obviously, which may be due to the good QD dispersion and ligand exchange.

Figure 4c shows 100 μm red epoxied QDCF pixel arrays. The patterns have excellent luminescence uniformity and durable adhesion. Generally, the clean edge of patterns demonstrates the homogeneous organic polymers and inorganic nanoparticles [20]. QDCF made of red epoxied QDs with toluene has good dispersion, surface smoothness, and compatibility, as shown in Figure 4c. By contrast, as shown in Figure 4d, a high concentration of pristine red QDCF shows many dark points within the pixel pattern. 

Figure 5 is pixelated green QDCF using epoxied QDs by photolithography. The single RGB pixel is 100 μm in size. Due to the presence of QDs, the polymerization rate decreased, which can be ascribed to the trapping of the initiator on the surface of QDs [21]. This phenomenon of imbalance of material components (Figure 4d) is suppressed by using epoxied QD. From the SEM images, we can clearly see the patterns are neat. It shows good dispersion, surface smoothness great compatibility.

## 4. Conclusions

In this study, we applied epoxy ligand exchange to obtain a good dispersibility of QD in a photoresist. In addition, 50 μm size QDCF arrays in a large area was obtained, which can be utilized in micro-LED application. In order to solve the problems of QD’s solubility in photoresist and chemical stability during the photolithography process, we demonstrated epoxy ligands modified QDCF. The QY of the QDCF remains almost unchanged after our photolithography process. Compared to the pristine, the red epoxy QDCF exhibited maximum QY up to 77.4%, with a high optical density over 1.2. Finally, we hope that the present work is a good starting point for understanding the properties of epoxied QDs and QDCFs. In the future, we intend to study QD with different diameters and materials related to the optical density enhancement of QDCFs.

## Figures and Tables

**Figure 1 micromachines-13-00595-f001:**
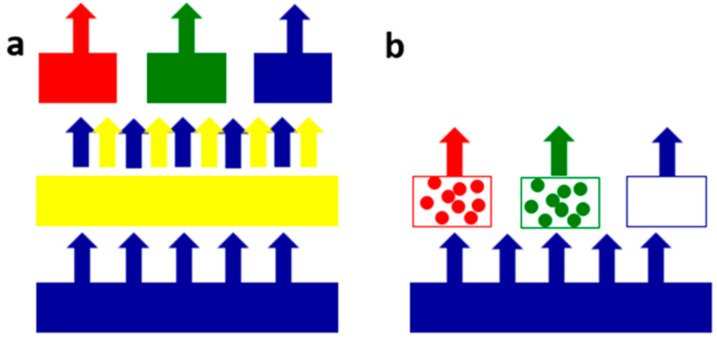
The structure of (**a**) traditional LCD backlight (**b**) QDCF excited by a blue backlight (Ex ≈ 450 nm).

**Figure 2 micromachines-13-00595-f002:**
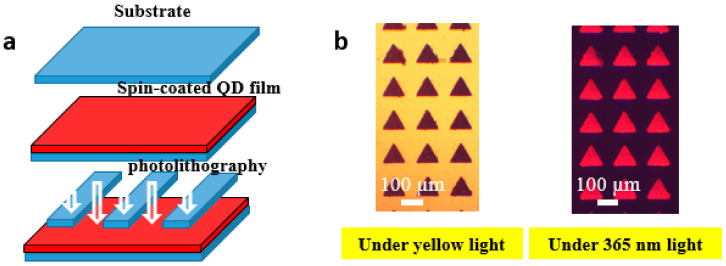
(**a**) The process flow of QDCF by photolithography; (**b**) the photolithography red QDCF after hard baking. Those triangles are the irradiated part, and the rest parts are the nonirradiated part.

**Figure 3 micromachines-13-00595-f003:**
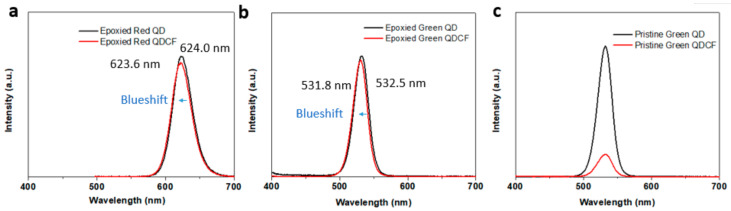
PL spectra (**a**) Epoxied red QD and QDCF; (**b**) Epoxied green QD and QDCF; (**c**) Pristine green QD and QDCF. The wavelength of the excitation light source is 450 nm.

**Figure 4 micromachines-13-00595-f004:**
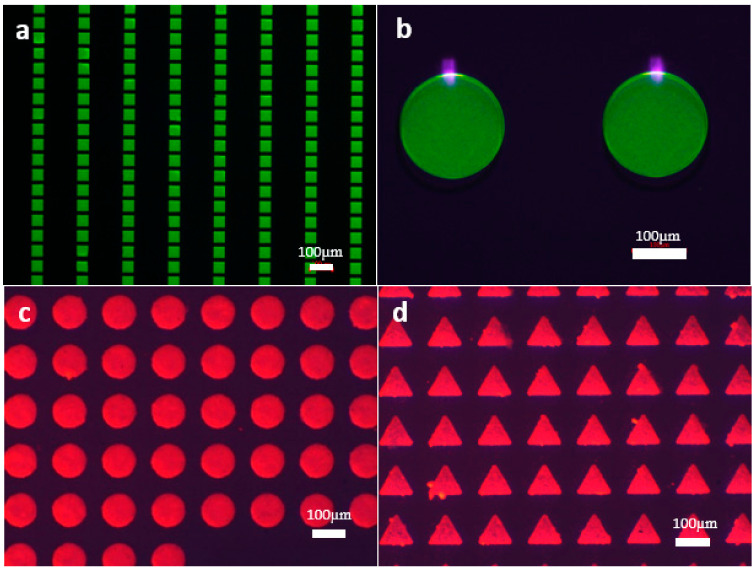
(**a**) 50 µm epoxied green QDs pixel in a large area; (**b**) epoxied green QDCF patterns with good uniformity; (**c**) 100 μm epoxied red QDCF pixel; (**d**) pristine red QDCF pixel. Blue light with a wavelength of 450 nm was used to excite QD films.

**Figure 5 micromachines-13-00595-f005:**
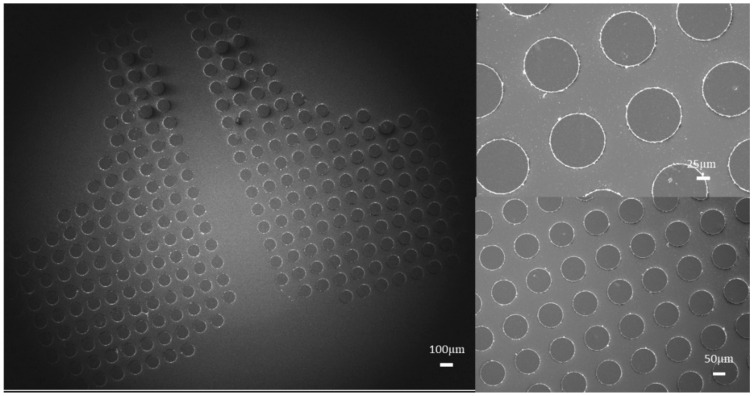
SEM images of the epoxied green QDCF with well-ordered homogeneous shapes.

**Table 1 micromachines-13-00595-t001:** Optical performance of QDs and QDCFs.

Parameter	Epoxied Red QDs	Epoxied Red QDCF	Epoxied Green QDs	Epoxied Green QDCF	Pristine Green QD	Pristine Green QDCF
QY of Film (%)	81.4	77.4	79.6	75.6	45.0	19.3
FWHM (nm)	32.5	33.8	25.0	25.8	25.0	26.6
Peak Wavelength (nm)	624.0	623.6	532.5	531.8	532.7	532.7

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
