# Peer review of "Red and Green Quantum Dot Color Filter for Full-Color Micro-LED Arrays"

_micromachines, 2022, doi:10.3390/mi13040595_

Round 1
Reviewer 1 Report
This paper reports on the fabrication of red and green quantum dot color filter with improved quantum yield using a unique ligand exchange process. I would like to recommend the publication of paper after some revisions.
1) Please give the wavelength of the excitation light source used for PL measurements.
2) What is the substrate used in the experiments? Is it an LED wafer?
3) Figure 2. It’s difficult for me to confirm the difference in uniformity and flatness of the QD films from this figure. What are the white bright areas (stripe shaped) appeared in these photos? More explanations are preferred.
4) The authors obtained an optical density of 1.2 for a QD film thickness of 6μm. However, this optical density is far from enough for micro-LED display applications, where a near-complete absorption of blue light is required. Please consider giving some discussions concerning how to realize a further larger optical density and/or how to further increase the QD film thickness.
5) The conclusion part, “XXX” seems to be a mistake.
Author Response
< Red and Green Quantum Dot Color Filter for Full-color Micro-LED Arrays>
Dear editor,
We have modified the manuscript (the Manuscript ID is 1614160) according to the suggestions made by the reviewers, which are marked in red in the annotated manuscript. We hope that the corrections will meet with the approval. The detailed corrections are listed below:
Reviewer: 1
Comments:
This paper reports on the fabrication of red and green quantum dot color filter with improved quantum yield using a unique ligand exchange process. I would like to recommend the publication of paper after some revisions.
1) Please give the wavelength of the excitation light source used for PL measurements.
Answer:
We thank the reviewer for the advices and we have modified our manuscript as below:
On page 2: the excitation light source - 150 W xenon light source was used for PL measurements. On page 4: The wavelength of the excitation light source is 450 nm.
2) What is the substrate used in the experiments? Is it an LED wafer?
Answer:
On page 2: the substrate used in the experiments is blue LED chip.
3) Figure 2. Itʼs difficult for me to confirm the difference in uniformity and flatness of the QD films from this figure. What are the white bright areas (stripe shaped) appeared in these photos? More explanations are preferred.
Answer:
The white bright areas are the reflection of room lamp. We thank the comment made by the reviewer and we have moved Fig. 2 from main text to Supplementary Information:
Figure S1. Chemical compatibility of different solvent for SU-8 photoresist.
4) The authors obtained an optical density of 1.2 for a QD film thickness of 6μm. However, this optical density is far from enough for micro-LED display applications, where a near-complete absorption of blue light is required. Please consider giving some discussions concerning how to realize a further larger optical density and/or how to further increase the QD film thickness.
Answer:
We agree with the comments made by the reviewer and we have rewrote our main text and conclusion as follows:
On page 3: “However, this OD is still not enough for micro-LED display applications. To realize a larger OD, an extra high QD concentration or larger film thickness is indispensable. Hence, we intend to further study QD-photoresist and hybrid structure in the future.”
On page 6: “Finally, we hope that the present work is a good starting point for understanding the properties of epoxied QDs and QDCFs. In the future, we intend to study QD with different diameters and materials related to the optical density enhancement of QDCFs.”
5) The conclusion part, “XXX” seems to be a mistake.
Answer:
We apologize for the mistake and we have now deleted “XXX” and edited the sentence in our manuscript.
We appreciate the instructive advice and kind help from you and the reviewers. The manuscript has been resubmitted to your journal, and hope that the correction will meet with the approval. Thank you very much for your considering our manuscript for potential publication.
Sincerely,
Bingxin Zhao

Reviewer 2 Report
This manuscript demonstrates a full-color display array by using color-conversion layers of red and green quantum dots color filter. By adjusting the ligand composition, the properties of the quantum dots and the pixel size of green QDCF were controlled to optimize the display performance. I think this work is interesting. However, there exist some obvious problems in this manuscript. It can be considered for publication after major revision.
- The description of prior research in the introduction somehow shallow. The author should cite more previous research and compare it with own work to highlight innovations;
- Page 2, Line 49-50. The description of the test equipment is flawed, and author should check the whole text carefully.
- Figure 2 needs to be embellished to differentiate more clearly, or put it into supplementary information directly.
- 3(a) is too simple, even without any notes, to show the process clearly. Authors need to consider making changes to it.
- Page 3, Line 94-96. “the existence of these epoxy groups facing the chemical and thermal treatment, which protect QDs from degradation.” Author need explain the specific mechanism or cite relevant literature to prove the protection mechanism.
- Page 3, Line 116. What meaning is “hard barking”? Baking?
- In figure 4, I think the PL peak position should be marked to show the blue shift or red shift.
- In Table 1, the data have no uniform format (significant number is inconsistent), which is not rigorous and non-standard. Author should correct it.
- I noticed that the scale in Figure. 5 is different in size. The scale in FIG. 5 (b) can be completely reduced to make it consistent with the other three figures, which is obviously better.
- Page 5, Line 158. “a ligand exchange method of XXX to obtain…” Author should check the manuscript carefully to avoid such ridiculous mistakes.
In addition, there are some grammatical errors and improper expression in this manuscript. Author should check and consider carefully to improve the quality of manuscript.
Author Response
< Red and Green Quantum Dot Color Filter for Full-color Micro-LED Arrays>
Dear editor,
We have modified the manuscript (the Manuscript ID is 1614160) according to the suggestions made by the reviewers, which are marked in red in the annotated manuscript. We hope that the corrections will meet with the approval. The detailed corrections are listed below:
Reviewer: 2
Comments:
This manuscript demonstrates a full-color display array by using color-conversion layers of red and green quantum dots color filter. By adjusting the ligand composition, the properties of the quantum dots and the pixel size of green QDCF were controlled to optimize the display performance. I think this work is interesting. However, there exist some obvious problems in this manuscript. It can be considered for publication after major revision.
- The description of prior research in the introduction somehow shallow. The author should cite more previous research and compare it with own work to highlight innovations;
Answer:
We thank the comment made by the reviewer, we subsequently rewrote our introduction which was inspired by the suggestion of the reviewer.
On page 1: “In preparing photolithography quantum dot color filters (QDCFs), fully cured cross-linking photoresist is proper to serve as QD matric for its excellent mechanical properties. However, QDs normally possess different surface chemical properties with photoresists, resulting in significant phase segregation when directly mixed QDs with a photoresist.[4]”
- Page 2, Line 49-50. The description of the test equipment is flawed, and author should check the whole text carefully.
Answer:
We thank the comment made by the reviewer and we have modified the description of the test equipment as follow:
2.2. Characterization
The PL spectra and quantum yield (QY) were recorded on C11347-11 Quantaurus-QY (Hamamatsu, Japan), the excitation light source - 150 W xenon light source was used for PL measurements. The substrate used in the experiments is blue LED chip. Photolithography was The brand of mask aligner is SUSS MicroTec (Germany). OD in this work was measured by Double-integrating-sphere system. Scanning electron microscopy (SEM) images were acquired with a field-emission scanning electron microscope (FE-SEM, JEOL JSM-7401F). All optical measurements were performed at RT under ambient conditions.
- Figure 2 needs to be embellished to differentiate more clearly, or put it into
supplementary information directly.
Answer:
We thank the comment made by the reviewer and we have moved Fig. 2 from main text to Supplementary Information:
Figure S1. Chemical compatibility of different solvent for SU-8 photoresist.
- 3(a) is too simple, even without any notes, to show the process clearly.
Authors need to consider making changes to it.
Answer:
We thank the comment made by the reviewer and we have modified as follow:
Figure 2. (a) The process flow of QDCF by photolithography; (b) the photolithography red QDCF after hard baking. Those triangles are Irradiated part, the rest parts are nonirradiated part.
- Page 3, Line 94-96. “the existence of these epoxy groups facing the chemical and thermal treatment, which protect QDs from degradation.” Author need explain the specific mechanism or cite relevant literature to prove the protection mechanism.
Answer:
We thank the comment made by the reviewer and we have added relevant literature as follow:
On page 3:
The surface of QD is covered with an epoxy ligand shell, the existence of these epoxy groups facing the chemical and thermal treatment, which protect QDs from degradation. [4]
[4] B. Zhao, X. Zhang, X. Bai, H. Yang, S. Li, J. Hao, H. Liu, R. Lu, B. Xu, L. Wang, K. Wang and X. W. Sun, Science China Materials, 2019, 62, 1463-1469.
- Page 3, Line 116. What meaning is “hard barking”? Baking?
Answer:
We apologize for the mistake and we have made the correction as “The QYs of different QDs and QDCF were compared, QDCFs are pixelated QD films after hard baking, as summarized in Table 1.”
- In figure 4, I think the PL peak position should be marked to show the blue
shift or red shift.
Answer:
We thank the comment made by the reviewer and we have modified Figure 3 as follow:
Figure 3. PL spectra (a) Green QDs; (b) Red QDs before and after ligand exchange with epoxy ligand. The wavelength of the excitation light source is 450 nm.
- In Table 1, the data have no uniform format (significant number is inconsistent), which is not rigorous and non-standard. Author should correct it.
Answer:
We thank the comment made by the reviewer and we have made the correction as follow:
Table 1. Optical performance of QDs and QDCFs.
Parameter |
Epoxied red QDs |
Epoxied red QDCF |
Epoxied green QDs |
Epoxied green QDCF |
Pristine green QD |
Pristine green QDCF |
QY of Film (%) |
81.4 |
77.4 |
79.6 |
75.6 |
45.0 |
19.3 |
FWHM (nm) |
32.5 |
33.8 |
25.0 |
25.8 |
25.0 |
26.6 |
Peak Wavelength (nm) |
624.0 |
623.6 |
532.5 |
531.8 |
532.7 |
532.7 |
- I noticed that the scale in Figure. 5 is different in size. The scale in FIG. 5 (b)
can be completely reduced to make it consistent with the other three figures, which is obviously better.
Answer:
We thank the comment made by the reviewer and we have modified the scale bar as follow:
Figure 5. SEM images of the epoxied green QDCF with well-ordered homogeneous shapes.
- Page 5, Line 158. “a ligand exchange method of XXX to obtain…” Author
should check the manuscript carefully to avoid such ridiculous mistakes.
Answer:
We apologize for the mistake and we have now deleted “XXX” and edited the sentence in our manuscript.
On page 5: “In this study, we applied epoxy ligand exchange to obtain a good dispersibility of QD in photoresist. In addition, 50 μm size QDCF arrays in large area was obtained, which can be utilized in micro-LED application.”
In addition, there are some grammatical errors and improper expression in this
manuscript. Author should check and consider carefully to improve the quality of
manuscript.
Answer:
We thank for the comments made by the reviewer, we checked whole manuscript and improved carefully.
We appreciate the instructive advice and kind help from you and the reviewers. The manuscript has been resubmitted to your journal, and hope that the correction will meet with the approval. Thank you very much for your considering our manuscript for potential publication.
Sincerely,
Bingxin Zhao

Round 2
Reviewer 2 Report
The author has answerd my questions and made some revisions. It can be accepted as this version.